# Decomposing Texture and Semantic for Out-of-distribution Detection

## Abstract

Out-of-distribution (OOD) detection tasks have made significant progress recently since the distribution mismatch between training and testing can severely deteriorate the reliability of AI systems. Nevertheless, the lack of precise interpretation for the in-distribution (ID) limits the application of the OOD detection methods to real-world systems. To tackle this, we decompose the definition of the ID into texture and semantics, motivated by the demands of real-world scenarios. We also design new benchmarks to measure the robustness that OOD detection methods should have. To achieve a good balance between the OOD detection performance and robustness, our method takes a divide-and-conquer approach. Specifically, the proposed model first handles each component of the texture and semantics separately and then fuses these later. This philosophy is empirically proven by a series of benchmarks including both the proposed and the conventional counterpart.

## 1 Introduction

Out-of-distribution (OOD) detection is the task that recognizes whether the given data comes from the distribution of training samples, also known as *in-distribution* (ID), or not. Any machine learning-based system could receive input samples that have a completely disparate distribution from the training environments (*e.g.,* dataset). Since the distribution shift can severely degrade the model performance (Amodei et al., 2016), it is a potential threat to reliable real-world AI systems.

However, the ambiguous definition of the ID limits the feasibility of the OOD detection method in real-world applications, considering the various OOD scenarios. For example, subtle corruption is a clear signal of OOD in the machine vision field while a change in semantic information might not be. On the other hand, an autonomous driving system may assume the ID from the semantic-oriented perspective; *e.g.,* an unseen traffic sign is OOD. Unfortunately, most of the conventional OOD detection methods and benchmarks (Zhang et al., 2021; Tack et al., 2020; Ren et al., 2019; Chan et al., 2021) assume the ID as a single-mode thus they cannot handle other aspects of OOD properly (Figure 1a).

To tackle this, we revisit the definition of the ID by decomposing it into two factors: *texture* and *semantics* (Figure 1b). For the texture factor, we define OOD as the textural difference between the ID and OOD datasets. On the contrary, the semantic OOD focuses on the class labels that do not exist in the ID environment. Note that the two aspects have a trade-off relationship, thus detecting both problems with a single model is challenging with the (conventional) entangled OOD perspective.

Geirhos et al. (2018) investigated the texture-shape cue conflict in the deep network and a series of subsequent studies (Hermann et al., 2019; Li et al., 2020; Ahmed & Courville, 2020) explored how to achieve a balance between these perspectives. However, they only analyzed the texture-shape bias inherited in deep networks. Instead, we focus on analyzing the texture and semantic characteristics underlying the ID to build a more practically applicable OOD detection method.

Unfortunately, to the best of our knowledge, none of the studies on the OOD detection benchmark have thoroughly analyzed the definition of the ID. It can be problematic when the method judges the image corrupted by negligible distortion as OOD, even when the environment can tolerate the small changes in texture. Because of such a complicated scenario, it is crucial to evaluate the OOD detection method in a comprehensive way that goes beyond the simple benchmark. Thus, in this study, we propose a new approach to measuring the performance of the method according to the decomposed

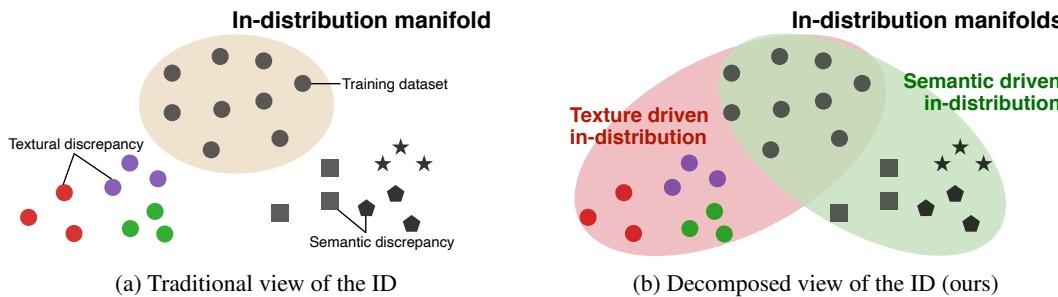

Figure 1: **How to define the ID? (a)** Traditional OOD detection studies manage the ID in an entangled view. However, this could be naïve considering the complex nature of the real environments. **(b)** We decompose the definition of the ID into texture and semantic; this provides the flexibility to handle complicated scenarios by determining which view of the ID is suitable for a given scenario.

definition of the ID. One notable observation in our benchmark is that most of the previous OOD detection methods are highly biased to the texture information and ignore the semantic clues.

To mitigate aforementioned issue, our proposed method tackles the texture and semantic information separately and aggregates these at the final module (Figure 2). To effectively extract texture information, we use a 2D Fourier transform motivated by the recent frequency domain-driven deep method (Xu et al., 2020). For the semantic feature, we design an extraction module upon the Deep support vector data description (Deep-SVDD) (Ruff et al., 2018) with a novel angular distance-based initialization strategy. We then combine two features using the normalizing flow-based method (Dinh et al., 2016), followed by the factor control mechanism. The control system provides the flexibility to handle different OOD scenarios by choosing which decomposed feature is more important in the given surrounding OOD circumstances. The main contributions of this work are as follows:

- We decompose the "unclear" definition of the ID into *texture* and *semantics*. To the best of our knowledge, this is the first attempt to clarify OOD itself in this field.

- Motivated by real-world problems, we create new OOD detection benchmark scenarios.

- We propose a novel OOD detection method that is effective on both texture and semantics as well as the conventional benchmark. Furthermore, our method does not require any auxiliary datasets or labels unlike the previous models.

## 2 RELATED WORK

**Class labels of the ID.** Early studies on deep OOD methods rely on class supervision. ODIN and Generalized ODIN (Liang et al., 2017; Hsu et al., 2020) use the uncertainty measure derived by the Softmax output. It determines a given sample as OOD when the output probability of all classes is less than a predefined threshold. (Sastry & Oore, 2020; Lee et al., 2018) utilize the extracted feature map (*e.g.,* gram matrix) from the pre-trained networks to calculate the OOD score. Also, Zhang et al. (2020) employ a flow-based model that is comparable to ours, but they require class information during the training and solely pay attention to semantics.

**Auxiliary distribution.** Outlier exposure (OE) (Hendrycks et al., 2018) exploits additional datasets that are disjoint from the test dataset to guide the network to better representations for OOD detection. (Papadopoulos et al., 2021) further improves the performance of OE by regularizing the network with the total variation distance of the Softmax output.

**Data augmentation.** Recently, contrastive learning-based methods have shown remarkable success on the tasks related to visual representation (He et al., 2020; Chen et al., 2020). Motivated by this, several studies employ data augmentation methods such as image transformation or additional noise on the OOD detection task or model (Hendrycks et al., 2019; Tack et al., 2020; Kirichenko et al., 2020).

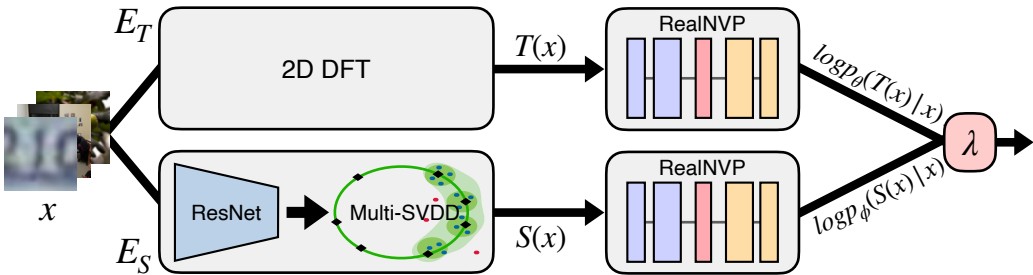

Figure 2: **Model overview.** Our framework extracts the texture and semantic information with the corresponding modules, then combines them via the normalizing flow-based method. **(a)** Texture feature $T(x)$ is distilled by the Fourier spectrum-based module. **(b)** We use multi-SVDD with a novel angular initialization to extract the semantic information $S(x)$. **(c)** Output features are merged by the explicit probability inference method, RealNVP. Here, we introduce the user control parameter $\lambda \in \{0.0, 0.5, 1.0\}$ to determine which feature is more suitable for a given OOD scenario.

Unlike the prior studies that exploit additional information other than the ID dataset, we only utilize the given (ID) training dataset. In addition, we separate and clarify the assumptions of OOD in terms of texture and semantics to improve the real-world practicability.

## 3 METHOD

We present an overview of the proposed method (Section 3.1), and each feature extraction module (Section 3.2 and 3.3). Finally, we introduce the normalizing flow-based conditional probabilistic modeling component (Section 3.4). Conventional OOD detection has an assumption that ID data are sampled from the distribution of the training dataset, $x \sim p_{data}$. We decompose the image with two factors and calculate the anomaly score based on each factor's likelihood. The texture module $T(x)$ uses Fourier analysis to extract frequency-based features from input $x$. The semantic branch $S(x)$ extracts the content label features such as shape. Our framework calculates the likelihood of these two factors and then combines them. Since we use the normalizing flow model that trains the exact likelihood, the extracted information is adjusted using a user-controllable hyperparameter $\lambda$.

### 3.1 MODEL OVERVIEW

We aim to train our method with the decomposed ID likelihoods, $p(T(x)|x)$ and $p(S(x)|x)$ as shown in Figure 2. With the given input image $x$, we extract features for each factor with different modules: texture- and semantic-aided sub components.

The extracted features are combined by the controllable normalizing flow method. Since the normalizing flow-based model explicitly calculates the negative log-likelihood, we model each extracted information as $\log p_\theta(T(x)|x)$ and $\log p_\phi(S(x)|x)$, where $\theta$ and $\phi$ are trainable parameters of the networks. In addition, we introduce the control parameter $\lambda \in [0.0, 0.5, 1.0]$ to model the final adjusted log-probability as $\lambda \cdot \log p(S(x)|x) + (1 - \lambda) \cdot \log p(T(x)|x)$. With this control mechanism, users can switch to the appropriate model "mode" by referring to their prior knowledge. For example, when the texture information overwhelms the semantic one for detecting OOD, we can overweight $\lambda$ for better performance. By default, we use the $\lambda$ value of 0.5 (no prior knowledge).

### 3.2 EXTRACTING THE SEMANTIC INFORMATION

**Multi-SVDD.** Beyond the one-class anomaly detection methods that consider the normal data as a single class (*e.g.,* DeepSVDD (Ruff et al., 2018)), recent studies have viewed the normal data as the union of the multiple hidden semantic information (Ghafoori & Leckie, 2020; Park et al., 2021). Inspired by this idea, we use the multi-SVDD method to extract the semantic information in an unsupervised manner for the OOD detection task.

Multi-SVDD embeds the samples to the multiple center vectors as closely as possible. Suppose a set of center vectors $\mathbf{C} = \{c_1, ..., c_K\}$ is initialized via K-means and the radius of each center is

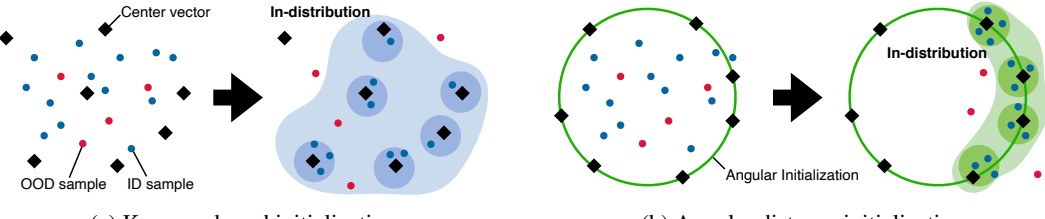

(a) K-means based initialization        (b) Angular distance initialization

Figure 3: **Comparison of the initialization strategy in multi-SVDD. (a)** The model with K-means initialization is effective at the anomaly detection but not for the OOD detection scenario. This is because the definition of the anomaly is when the samples do not belong to the cluster regions (dark shade), while the definition of OOD is when the samples do not lie in the ID manifold (light shade). **(b)** Our proposed angular distance-based initialization guides the initial center vectors to be positioned following a (virtual) circular line. As a result, it prevents the OOD samples from not belonging to the ID by creating tight cluster layouts without a hole.

$r = [r_1, ..., r_K]$. In multi-SVDD, the objective function is defined as follows.

$$\min_{\mathcal{W}, \mathbf{r}} \quad \sum_{k=1}^{K} r_k^2 + \frac{1}{\nu n} \sum_{i=1}^{n} \max\left\{0, \|\phi(x_i; \mathcal{W}) - c_j\|^2 - r_j^2\right\} + \frac{\eta}{2} \sum \|\mathcal{W}\|^2. \tag{1}$$

Here, $\phi(x_i; \mathcal{W})$ is the deep network with a set of weight parameters $\mathcal{W}$ and $c_j$ is assigned to $\phi(x_i; \mathcal{W})$. As the set $r$ is decreased, the samples are condensed into the center vectors. By using the distance between the center vectors and the samples, we get an anomaly score.

**Angular distance initialization.** SVDD is proposed for the anomaly detection task. Because of the disparity between OOD and anomaly detection, the direct application of the SVDD-based model to OOD detection causes unexpected performance degradation. In anomaly detection, although the abnormal samples lie in the ID manifold, it is possible to detect them as abnormal unless they are close to the center vectors $c_j$. For example, as in Figure 3a, OOD samples (red) that are located inside of the ID manifold (light blue shade) can be detected as abnormal since they are outside of the tight cluster boundary (dark blue shade). Because of these, a mixture of Gaussian probability density is a reasonable density space for the anomaly detection model.

Unlike the anomaly detection task, the OOD detection is to find the samples that are not ID. In Figure 3a, all the OOD samples placed in the ID manifold (light dark shade) are recognized as the ID. We propose an angular distance-based center vector initialization strategy to tackle this issue. Angular-based embedding has a clear geometric interpretation (Deng et al., 2019), and they can handle the complex and subtle differences in feature space. Our proposed method is as follows:

$$c_k = \gamma \frac{\mathbf{v}}{\|\mathbf{v}\|}, \quad \mathbf{v} \in \mathbb{R}^h \sim \mathcal{N}(\mathbf{0}, \mathbf{1}) \tag{2}$$

where $h$ is the dimension of the embedding space and $\gamma$ is the hyper-parameter for the radius of the sphere. After $\phi(x_i; \mathcal{W})$ is trained based on angular initialization, semantic features are extracted through this model; $S(x_i) = \phi(x_i; \mathcal{W})$. By setting the $\gamma$ value sufficiently large, we ensure that all sample data are within a radius of the sphere as illustrated in Figure 3b. While Equation 1 drives the training samples to be embedded around the center vectors on the sphere, the OOD samples remain near the origin. This embedding space may be weak to recognize the semantic label of a given sample, but it is sufficient to identify whether a sample is OOD or not.

### 3.3 EXTRACTING THE TEXTURE INFORMATION

To effectively extract the texture property of the ID, we interpret the image in the frequency space. With a given input image $x \in \mathbb{R}^{3 \times h \times w}$, we first convert it into the frequency domain using Discrete Fourier Transform (DFT) $\mathcal{F}$ as shown below.

$$\mathcal{F}(f_x, f_y) = \frac{1}{hw} \sum_{p=0}^{h-1} \sum_{q=0}^{w-1} I(p, q) \cdot e^{-i2\pi(f_x p/h + f_y q/w)}, \tag{3}$$

Here, $I(p,q)$ denotes the pixel value of the image at the $(p,q)$-coordinate and $\mathcal{F}(f_x,f_y)$ is the output of the DFT at the Cartesian coordinate $(f_x,f_y)$ in the frequency space. In order to construct a scale and rotation invariant frequency information in 2D image, we modify the coordinate system from Cartesian $(f_x,f_y)$ to polar $(f_r,\theta)$, following (Dzanic et al., 2019).

$$\mathcal{F}(f_r,\theta) = \mathcal{F}(f_x,f_y) \quad : \quad f_r = \sqrt{\frac{f_x^2 + f_y^2}{\frac{1}{4}(m^2 + n^2)}}, \quad \theta = \operatorname{atan2}(f_y, f_x). \tag{4}$$

Since directly computing the polar coordinate is computationally expensive and tricky, we iteratively calculate the rotation invariant frequency feature. To do that, we only utilize the first channel of the image as $x \in \mathbb{R}^{1 \times h \times w}$ and assume that the image is square (*i.e.*, $h = w$). Let $T(x) \in \mathbb{R}^{w/2}$ be the texture feature vector of the image $x$. Then, the $i$-th element of $T_i(x)$ is calculated by Equation 5.

$$T_i(x) = R_i - R_{i-1} \quad : \quad R_w = \sum_{f_x=-w}^{w} \sum_{f_y=-w}^{w} \mathcal{F}(f_x, f_y), \quad R_0 = \mathcal{F}(0,0) \tag{5}$$

Note that $f_r$ represents the direction of the frequency of an image. Since $T_i(x)$ gathers the boundary region of the rectangular-shape feature map, $R_i$ (as Equation 5), the same frequency power of an image collapses into $T_i(x)$. Consequently, $T(x)$ loses the information of the object's shape and it only contains the frequency-based texture information.

**Discussion.** We compare the power spectrum density (PSD) of CIFAR-10 (C10), CIFAR-100 (C100), and distorted C10 (Figure 4). The PSD discrepancy between the corrupted C10 and the vanilla one indicates that the image feature acquired from the frequency domain is adequate to represent the texture cue. In contrast, C10 and C100 are not distinguishable in the frequency domain since they have a very similar image texture due to the small resolution. These observations support our assumption that the OOD detection model should decouple the texture and semantic information to handle real-world cases properly.

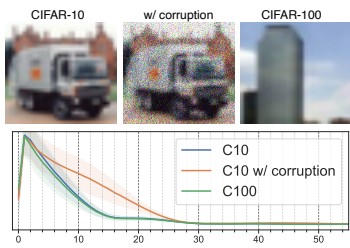

Figure 4: **PSD analysis.**

### 3.4 FEATURE COMPOSITION VIA NORMALIZING FLOW

Since we design our framework to directly sample the probability, it does not require any ad-hoc scoring functions. Instead, we use a normalizing flow-based method (Dinh et al., 2014; Rezende & Mohamed, 2015; Dinh et al., 2016) that uses the probability of given samples as a loss function. In the following, we will describe how to get the probability of samples from the prior probability (Normal distribution) using the normalizing flow.

Given sample $x$, a normal prior probability distribution $p_Z$ on a latent variable $z \in Z$, and a bijection $f : X \to Z$ (with $g = f^{-1}$), the change of variables defines a model distribution on $X$ by

$$p_X(x) = p_Z\big(f(x)\big) \left| \det\left(\frac{\partial f(x)}{\partial x^T}\right) \right|, \tag{6}$$

where $\frac{\partial f(x)}{\partial x^T}$ is the Jacobian of $f$ at $x$. The function $f$ can be decomposed into $f = f_1 \circ \cdots \circ f_k$. To compose the features, we use a flow-based RealNVP with coupling layers (Dinh et al., 2016). Here, $T(x_i)$ and $S(x_i)$ are input of RealNVP. Given the ID training dataset $\mathcal{D}$, the objective of the RealNVP model with the trainable parameters $\theta$ and $\phi$ is to minimize the following loss function.

$$\mathcal{L}_t(\mathcal{D}) = \frac{1}{N} \sum_{i=1}^{N} -\log p_\theta(T(x_i)|x_i), \quad \mathcal{L}_s(\mathcal{D}) = \frac{1}{N} \sum_{i=1}^{N} -\log p_\phi(S(x_i)|x_i) \tag{7}$$

Although we extract image features from two different aspects, it is not guaranteed that the features are completely disentangled. To mitigate the unexpected entangling, we propose disentanglement loss using the independence property of log probabilities. If $T(x_i)$ and $S(x_i)$ are independent given $x_i$, then $\log p(T(x_i)) + \log p(S(x_i)) = \log p(T(x_i), S(x_i))$. Note that $\log p(T(x_i), S(x_i))$ can be computed by concatenating features $S(x)$ and $T(x)$. The disentanglement loss is calculated as:

$$\mathcal{L}_{disentangle} = \frac{1}{N} \sum_{i=1}^{N} \{(\log p(T(x_i)) + \log p(S(x_i))) - \log p(T(x_i), S(x_i))\}^2. \tag{8}$$

| ID $\rightarrow$ OOD | | Using Labels | | Other Dist. | | Self-supervised | | | Ours $\lambda =$ | | |
|---|---|---|---|---|---|---|---|---|---|---|---|
| | | Maha | Gram | OE | OEC | Rot | SSL | CSI | 0.0 | 0.5 | 1.0 |
| SVHN | C10 | 99.3 | 97.3 | 99.3 | 99.8 | - | 99.8 | - | 93.9 | 99.9 | **100.** |
| | C100 | - | - | 99.0 | **99.9** | - | 99.8 | - | 91.1 | 99.8 | **99.9** |
| | TinyImgNet | 99.3 | 97.3 | - | - | - | - | - | 99.5 | **100.** | **100.** |
| | LSUN* | 99.9 | 99.8 | 99.9 | 99.9 | - | 99.9 | - | 99.9 | **100.** | **100.** |
| C10 | SVHN | 99.1 | 99.5 | 98.2 | 99.2 | 97.8 | 99.2 | 99.8 | 86.1 | **99.9** | **99.9** |
| | C100 | 88.2 | 79.0 | 92.9 | **93.8** | 82.3 | 93.8 | 89.2 | 55.7 | 93.6 | 93.5 |
| | TinyImgNet | 99.5 | 99.7 | - | - | - | - | - | 65.4 | **99.9** | **99.9** |
| | LSUN* | 99.3 | **99.9** | 96.4 | 98.9 | 92.8 | 98.9 | 97.5 | **99.9** | 98.8 | 81.9 |
| C100 | SVHN | 98.4 | 97.3 | 82.8 | 95.8 | - | 95.8 | - | 80.0 | 99.9 | **100.** |
| | C10 | 77.5 | 67.9 | 77.5 | 77.7 | - | 77.7 | - | 49.4 | 83.1 | **84.2** |
| | TinyImgNet | 97.4 | 99.0 | - | - | - | - | - | 91.7 | **100.** | **100.** |
| | LSUN* | 98.2 | 99.3 | 79.5 | 88.8 | - | 88.8 | - | 99.9 | **100.** | **100.** |

Table 1: **Conventional OOD detection benchmark.** We evaluate the detection performance by AUC (in- vs. out-distribution detection based on confidence/score) in percentage (**higher is better**). * indicates the high-resolution dataset.

The final loss is the sum of all the aforementioned losses; $\mathcal{L} = \mathcal{L}_t + \mathcal{L}_s + \mathcal{L}_{disentangle}$. One nice side-effect of this probabilistic modeling for texture-semantics decomposition is that we can adjust the contribution of these components by referring to the given OOD environment. Since we designed the features to be separate, we combine them with a linear interpolation as:

$$\lambda \cdot \log p_\phi(S(x)|x) + (1 - \lambda) \cdot \log p_\theta(T(x)|x) \tag{9}$$

where $\lambda \in [0.0, 0.5, 1.0]$ is the control parameter (default is 0.5). In our experiments, we denote $\lambda = 0.0$ as "texture mode" and $\lambda = 1.0$ as "semantic mode".

## 4 EXPERIMENTS

We demonstrate the effectiveness of our method on various benchmark setups. In Section 4.1, we report the performance on the conventional OOD detection task and discuss the limitations of the previous OOD detection studies. We use the area under the curve (AUC) for the receiver operating characteristic (ROC) curve to evaluate the performance. Then, we will discuss the factor-aware OOD detection that is proposed to fill the gap between the current benchmark and the real-world environment (Section 4.2). We conclude this part with model analysis (Section 4.3).

### 4.1 CONVENTIONAL OOD DETECTION

**Setups.** Here, we evaluate OOD detection methods on the widely used OOD detection benchmark. We select SVHN (Netzer et al., 2011), CIFAR-10 (Krizhevsky et al., 2009) (C10), and CIFAR-100 (Krizhevsky et al., 2010) (C100) as the ID dataset. To simulate the OOD samples, LSUN (Yu et al., 2015) and Tiny-ImageNet (Torralba et al., 2008) datasets are additionally used.

**Baselines.** We compare our approach to the methods belonging to three different OOD detection groups. **1)** Methods that use class labels of the training samples; feature-based methods such as Maha (Lee et al., 2018) and Gram (Sastry & Oore, 2020) fall into this category. **2)** Methods that utilize an additional distribution (dataset) such as OE (Hendrycks et al., 2018) and OEC (Papadopoulos et al., 2021). **3)** Self-supervised based methods; rotation-based (Rot) (Hendrycks et al., 2019), SSL (Mohseni et al., 2020), and CSI (Tack et al., 2020) are in this group.

**Results.** As shown in Table 1, our proposed method surpasses the competitors on the conventional OOD detection tasks without using any extra information such as class labels, other datasets, or image-transformation techniques that the other methods required. In detail, ours with $\lambda = 1.0$ (semantic mode) achieves the best performance in all the cases with the exception of two scenarios.

**Discussion.** As we have discussed, it is often risky to assume that the input samples of a real-world system have a single and entangled characteristic. We argue that it is more natural to consider in

terms of multiple factors such as texture and semantics when detecting OOD. For example, when a given environment requires detection of the textural discrepancy as OOD, then the detection method should concentrate on the textural side alone, not the semantic counterpart.

Unfortunately, conventional OOD detection benchmarks are not developed to measure the disentangled view of the ID. Here, we dissect the traditional benchmark to quantify the effect of each factor we decompose. We first categorized the datasets into the high- and low-resolution groups by their image resolution: C10, C100, SVHN, and TinyImageNet belong to the low-resolution (LR) group while LSUN is high-resolution (HR). Table 1 shows that previous studies achieve high performance on the LR → HR scenario, and especially the label-based methods produce outstanding results. However, we will show the superior performance of these methods derives from their abuse of texture information (*e.g.,* detect as OOD by highly referring to image resolution) in Section 4.2.

On the other hand, we argue that the semantic property is the key component to identity the OOD of LR → LR scenarios. For example, C10 ↔ C100 solely requires semantic information to detect OOD since these datasets use very similar images (in terms of the texture and image resolution). This is the reason why ours with $\lambda = 0.0$ (texture mode) cannot detect OOD at all (55.7 and 49.4 AUC). Note that other competitors also show inferior performance, especially in the C10 ↔ C100 case, which demonstrates that these methods have weakness at handling semantic information.

## 4.2 FACTOR-AWARE OOD DETECTION

**Motivation.** For a more in-depth evaluation of OOD detection performance, we propose a novel OOD detection benchmark in which multiple factors (or environments) are involved. Conventional setups simply define the ID and OOD as different datasets. Unfortunately, this assumption is often invalid in real-world use cases. Imagine a distorted image scenario; would we have to label this corrupted image as OOD? Most previous studies would obviously determine it as OOD, but this heavily depends on the surrounding circumstances; for example, if an OOD detection system is granted a role to send an alert when a new class object appears, then it has to disregard mild corruption.

Therefore, the goal of this benchmark is to evaluate from the *OOD detection performance* (Figure 5a and b) and the *robustness*, whether the model can judge the given sample as ID when it differs from the training set while still coming from inside the "interest boundary" (Figure 5c and d). One consecutive question may arise: what kind of factors should be considered? Certainly, in real-world applications, various factors can be derived from given samples or datasets. However, we narrow the scope of the factors to texture (Figure 5a and c) and semantics (Figure 5b and d), since these are the most important aspects considering the decision-making process in the real world.

**Setups.** To evaluate OOD detection performance, AUC is used; 100.0% AUC is the best. To measure robustness, we define 50% AUC as the best since this indicates that the detection method cannot distinguish test samples as OOD (determined as ID). In this benchmark, we use corrupted C10, SVIRO (Cruz et al., 2020), BTAD (Mishra et al., 2021), MNIST variants, and rotated MNIST datasets as shown in Figure A.1. Detailed descriptions of each dataset are described in the experience parts.

**Detecting texture discrepancy of OOD.** The goal of this scenario is to detect the texture discrepancy. In real-world application, the factory process requires quality control to find cracks in industrial products. To meet the demand, we use BTAD dataset (Mishra et al., 2021), which is composed of 2,830 industry-oriented images of normal and defective products. We set the normal samples as the ID and the cracked ones as OOD. Furthermore, as discussed earlier, we simulate the severe distortion scenario by applying severe levels of frost, shot noise, haze, and motion blur to C10. In this case, the model is required to detect the distortions as OOD, simulating the circumstances where the system should find the corrupted sensing module.

As shown in Table 2, our method with texture mode ($\lambda = 0.0$) outperforms all the others in BTAD case and achieves comparable performance for severe distortion scenario. On the other hand, the semantic mode ($\lambda = 1.0$) shows substantially lower AUC. This is unsurprising because the current benchmark scenario does not require semantic information; distortions and cracks (BTAD) are only related to the texture clue. Gram and Maha also achieve high performance; however, we find that these methods do not have sufficient robustness as will be discussed in the Robustness experiment.

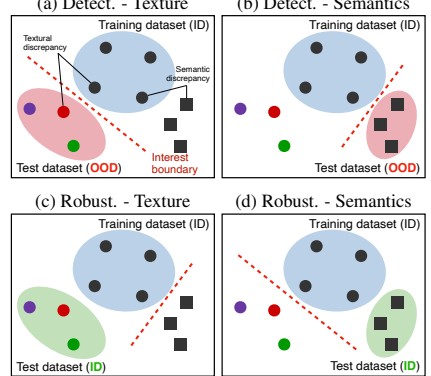

| Goal - Factor | AUC (Oracle) | Experiments |
|---|---|---|
| Detect - Texture | 100. | BTAD (Mishra et al., 2021) (T2) |
| | | Severe distortion (T2) |
| Detect - Seman. | | Rotated MNIST (T3) |
| | | MNIST↔F-MNIST (T3) |
| Robust - Texture | 50.0 | Image resolution (T4) |
| | | Mild distortion (T4) |
| Robust - Seman. | | SVIRO (Cruz et al., 2020) (T5) |
| | | MNIST↔K-MNIST (T5) |

Figure 5: **Taxonomy of our proposed benchmark. Left:** (a, b) OOD detection scenario on two factors: texture, and semantics. In this case, the training set is ID, while the test set is OOD. (c, d) Evaluating the robustness on factor discrepancy of ID by providing ID samples at test time. Here, an oracle produces 50.0 AUC (cannot distinguish it as OOD at all) since the given samples are from ID. **Right:** A list of the corresponding experiments in each scenario.

| Method | Severe distortion C10→ | | | | BTAD Normal→ |
|---|---|---|---|---|---|
| | Frost | Haze | Motion | Shot | Anomaly |
| ODIN | 79.9 | 69.5 | 80.9 | 84.2 | 52.2 |
| Maha | 97.2 | 99.8 | 97.4 | 99.8 | 79.1 |
| Gram | **99.9** | **99.9** | **99.9** | 99.8 | 81.1 |
| CSI | 84.3 | 81.6 | 87.6 | 97.1 | 61.1 |
| 0.0 | 83.4 | 94.4 | 91.0 | 96.1 | **92.1** |
| $\lambda = 0.5$ | 80.4 | 73.5 | 90.8 | 82.5 | 90.0 |
| 1.0 | 67.5 | 67.1 | 63.1 | 74.1 | 61.0 |

Table 2: Comparison of the texture discrepancy detection of OOD.

| Method | Rotated MNIST 0°→ | | | | | MNISTs MNIST→ |
|---|---|---|---|---|---|---|
| | 15° | 30° | 45° | 60° | 75° | FMNIST |
| ODIN | 68.1 | 84.4 | 91.9 | 96.3 | 97.7 | 74.4 |
| Maha | 80.0 | 86.1 | 88.1 | 87.9 | 89.9 | 96.8 |
| Gram | **88.8** | **91.0** | 96.6 | **98.8** | 99.1 | **99.9** |
| CSI | 59.2 | 60.0 | 60.8 | 61.8 | 61.9 | 56.1 |
| 0.0 | 51.1 | 51.1 | 52.3 | 54.6 | 59.1 | 91.8 |
| $\lambda = 0.5$ | 65.5 | 84.1 | 90.1 | 95.5. | 91.1 | 100. |
| 1.0 | 67.1 | 89.6 | **96.8** | 98.5 | **99.8** | **99.9.** |

Table 3: Comparison of the semantic discrepancy detection of OOD.

**Detecting semantic discrepancy of OOD.** With this benchmark, we evaluate the method's sensitivity to the semantic difference between the training and test samples. Catching semantic discrepancies is important in many applications such as autonomous systems. For example, this should judge whether a given traffic symbol is geometrically distorted for a stable recognition system (some recognition models are vulnerable to rotation). Here, we use two testbeds: MNIST→F-MNIST and rotated MNIST scenarios. For the rotated MNIST, we train the model on MNIST and evaluate the rotated version. In most cases, our method with semantic mode surpasses the others (Table 3).

**Robustness on Texture Discrepancy of ID.** We conduct the benchmark on two scenarios: mild distortion and varying image resolution. Both cases assume that the definition of OOD is in the disparity of semantic information. Particularly, a subtle change of image should not be considered as a signal of OOD, but rather recognized as ID. For example, the system that is requested to detect abnormal events should be robust on the mild image distortions. To implement the distortion case, we corrupt C10 with a mild level. For the varying resolution, we center-crop and resize back to the original resolution. We carefully adjust the crop operation so as not to harm the image information.

Table 4 shows the AUC score of this benchmark. Since this experiment focuses on detecting the test samples as the ID, 50.0% AUC is the best. We observe that the deep feature-based methods (*e.g.,* Maha and Gram) have little robustness against the texture discrepancies. They are not potentially appropriate for applications that do not treat a mild corruption as OOD. Furthermore, all the methods except ours with semantic mode are extremely sensitive to image resolution change, although no other information is modified. In contrast, our method with $\lambda = 1.0$ (semantic mode) achieves the best performance because this focuses on the semantic information alone.

**Robustness on Semantic Discrepancy of ID.** Unlike the above, we focus on the robustness against the semantic discrepancy presented in ID. To evaluate this, we use MNIST→KMNIST and

| Method | Mild distortion | | | | Image resolution | | | |
|---|---|---|---|---|---|---|---|---|
| | C10→ | | | | $32^2 \to$ | | | |
| | Frost | Haze | Motion | Shot | $36^2$ | $40^2$ | $44^2$ | $48^2$ |
| ODIN | 72.6 | 75.2 | 80.9 | 84.2 | 55.7 | 62.7 | 67.5 | 91.2 |
| Maha | 88.3 | 99.3 | 97.4 | 99.8 | 69.0 | 71.1 | 79.9 | 80.0 |
| Gram | 99.8 | 99.6 | 99.9 | 99.8 | 66.5 | 74.8 | 82.4 | 92.9 |
| CSI | 73.1 | 60.1 | 87.6 | 97.1 | 81.5 | 86.1 | 89.9 | 93.5 |
| 0.0 | 68.8 | 59.3 | 79.0 | 91.1 | 59.9 | 67.7 | 75.8 | 82.8 |
| $\lambda = 0.5$ | 64.3 | 58.0 | 71.8 | 82.5 | 50.2 | 55.8 | 71.2 | 72.8 |
| 1.0 | **56.6** | **55.0** | **54.1** | **74.1** | **50.1** | **55.0** | **60.9** | **66.0** |

Table 4: Comparison of the robustness on the texture discrepancy of ID.

| Method | SVIRO | | | | MNISTs |
|---|---|---|---|---|---|
| | A-class→ | | | | MNIST→ |
| | Tesla | Escape | Tiguan | i3 | KMNIST |
| ODIN | 67.3 | **53.4** | 56.2 | 58.9 | 94.4 |
| Maha | 90.0 | 76.0 | 88.1 | 77.2 | 91.6 |
| Gram | 91.1 | 91.9 | 81.1 | 81.2 | 97.1 |
| CSI | 65.5 | 54.4 | 53.2 | 57.2 | 72.2 |
| 0.0 | **60.0** | 59.4 | 55.8 | **55.3** | **50.6** |
| $\lambda = 0.5$ | 90.0 | 79.2 | 79.0 | 76.6 | 100. |
| 1.0 | 96.5 | 84.1 | 87.6 | 90.9 | 100. |

Table 5: Comparison of the robustness on the semantic discrepancy of ID.

| Method | C10→ | | C100→ | | SVHN→ | |
|---|---|---|---|---|---|---|
| | C100 | SVHN | C10 | SVHN | C10 | C100 |
| Multi-SVDD | 53.7 | 99.7 | 67.5 | 84.2 | 87.5 | 84.2 |
| + $L_{disentangle}$ | 69.5 | 99.8 | 70.9 | 91.0 | 99.9 | 98.9 |
| + Angular init. | **93.5** | **99.9** | **84.2** | **100.** | **100.** | **99.9** |

Table 6: **Ablation study on the semantic module.** Multi-SVDD without our proposed components cannot properly handle the semantics information.

| Method | C10 | C100 | SVHN |
|---|---|---|---|
| ODIN | 56.2 | 53.3 | 52.2 |
| Gram | 79.1 | 69.1 | 71.1 |
| CSI | 61.9 | 52.2 | 51.1 |
| Ours | **50.1** | **49.8** | **50.0** |

Table 7: **ID→ID scenario.** All the models are trained on the "training set" of the ID dataset and evaluated on its "test set".

SVIRO (Cruz et al., 2020) scenarios. SVIRO is the collection of images from diverse vehicle interior rear seats. Among them, we set Mercedes A-class as ID and other car brands as the OOD datasets. With this, we simulate the following scenario: the machine learning system utilizing the camera installed inside of the vehicle should be robust to the vehicle type (semantics discrepancy).

For MNIST → K-MNIST (Table 5), our method with $\lambda = 0.0$ (texture mode) successfully determines that K-MNIST is the ID (AUC is 50.6). In contrast, our model with other $\lambda$ values (0.5 and 1.0) cannot, since these mostly depend on the semantic clue which is meaningless in this case. Similarly, our method with $\lambda = 0.0$ (texture mode) achieves good performance on SIVRO.

### 4.3 MODEL ANALYSIS

**Semantic module.** We validate the proposed component of the semantic module (Table 6). The disentangle loss plays an important role in the model performance and the angular initialization also improves the conventional initialization method in a huge margin, implying the superiority of our method when extracting the semantic information. To further present the effectiveness of our strategy, we plot the distribution of the embedding results in Figure B.2.

**ID → ID scenario.** If we test an OOD detection model using the test set of the ID dataset, then ideally the prediction for it should be ID. On the other hand, when the OOD detection model overfits the training set, then the AUC score for the test set would go to 100.0%. As shown in Table 7, our method determines the ID almost perfectly, and CSI and ODIN also show reasonable performance. However, the Gram-based method is rather sensitive to this and we argue that it might be due to abuse of features of the seen (training) dataset.

## 5 CONCLUSION

In this work, we introduce a novel viewpoint of the ID for the practically applicable OOD detection. We separate the definition of the "single-mode" ID into "texture" and "semantics" factors by following the requirements of the real-world applications. To effectively handle both aspects, we take a divide-and-conquer strategy that extracts the features using the appropriate method in each factor and then combines these with the normalizing flow-based model. By doing so, our method outperforms previous models on both our newly proposed benchmark scenarios and the conventional OOD detection cases. We hope that our work can provide useful guidance for future OOD detection work.

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

# A EXPERIMENT SETUPS

## A.1 MODEL TRAINING

We use Adam (Kingma & Ba, 2014) with a weight decay of 5e-5 and a batch size of 256 for 100 epochs to train all the modules in our framework. Extracted features by the $E_T$ and $E_S$ have the same 112 dimensions. We train and test with a single NVIDIA 1080TI GPU.

## A.2 MODEL ARCHITECTURE

**Multi-SVDD.** We use ResNet-18 as the encoder network and modify the dimension of the last layer to 512. All the hidden dimensions of semantic modules have the same as the original setup. The $\gamma$ parameter for the angular distance is set as 250 for all in-distribution datasets.

**RealNVP.** We use the RealNVP implementation following (Izmailov et al., 2020)[1]. In detail, each perspective model has 8 blocks of 8 flows and we use an affine coupling that is defined by the fully connected shift and scale networks each of which 16 dimensions of hidden layers.

## A.3 OOD BENCHMARK

**Datasets.** Figure A.1 shows examples of the datasets in the proposed OOD detection benchmark. To quantify the role of texture information, we use corrupted C10 dataset (Figure A.1a). We distort images of the C10 dataset with frost, haze, motion blur, and shot noises (following C10-C settings (Hendrycks & Dietterich, 2019)). With such corrupted C10 scenarios, we try to simulate the weather change or small sensing error of the vision system. Note that we separate the distortion scenario into mild and severe corruption cases to cover the various demands from the real applications. Specifically, the mild scenario is utilized to evaluate the textural robustness of the method. It simulates the environments where slight textural deviation should be ignored and the algorithm should concentrate on the semantics changes. For instance, the vision-based surveillance system requires such capability since it should be robust to minor changes (*e.g.,* weather, dust, etc.) and detect the environment change regarding the semantics (*e.g.,* trespassing). The severe distortion case, on the other hand, is used to measure the OOD detection performance with a given textural change. A system that should notice the deviation of textural dissimilarity falls in this category.

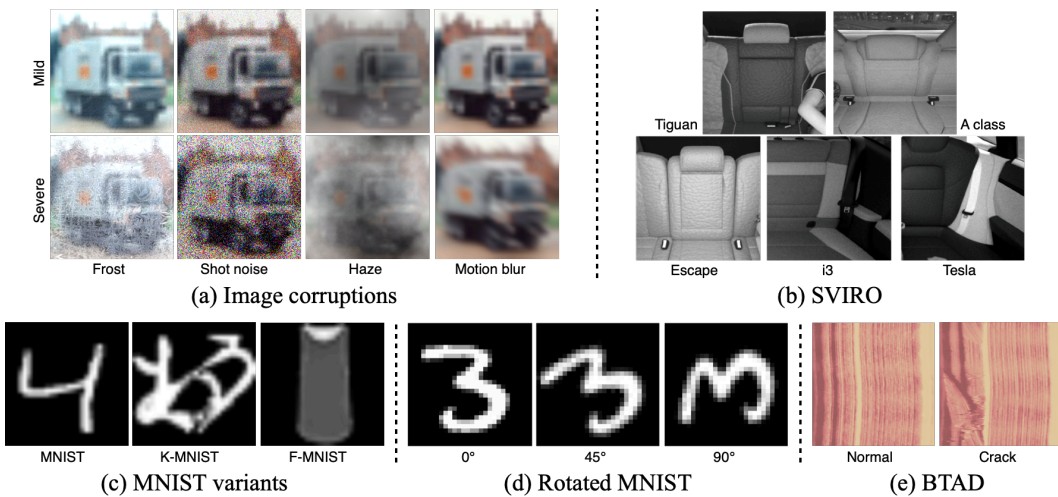

Figure A.1: **Examples of the datasets used in the proposed benchmark.** The corresponding experiment for each dataset is illustrated in Figure 5.

Figure A.1b displays the SVIRO dataset (Cruz et al., 2020), which is a collection of vehicle interior images. Each car dataset is consist of images taken on the rear seats with six car brands (as shown in figure) and among them, we set the ID as whole directions of Mercedes A-class rear seat and OOD

[1]https://github.com/izmailovpavel/flowgmm

as other car datasets. With this dataset, we measure the robustness of the semantics change for the following potential scenario: The machine learning system utilizing the camera installed inside of the vehicle should not behave differently although the vehicle interior is changed. In other words, the system should be robust to the vehicle type (semantics discrepancy).

MNIST variants such as K-MNIST and F-MNIST are used to measure semantics-aware benchmarks (Figure A.1c). In addition, we generate a rotated MNIST dataset by rotating images of the MNIST dataset as illustrated in Figure A.1d. Since MNIST-based datasets are composed of grayscale images, we replicate the color channel by three times before feeding into the detection method.

To evaluate the ability to detect the products' crack from the real environment, we use the BTAD dataset (Mishra et al., 2021) as shown in Figure A.1e. In detail, we selected the cracked image from the test dataset of BTAD and reorganize it as the ID dataset contains only non-cracked (normal) product images and the OOD dataset has cracked images only.

**Image resolution.** When we are able to acquire the official pre-trained network or samples, we provide the $32\times32$ resolution images as input, which is de facto setup in this field. If not the case (such as our benchmark), we prepossess the image as to have $32\times32$ resolution following the official setup. In our method, except for the resolution change experiment, we resize the images to $112\times112$ resolution to give enough information to the texture extraction module.

## B  MODEL ANALYSIS

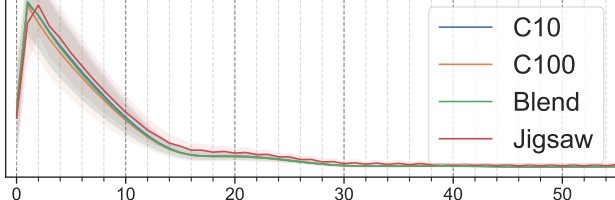

Figure B.1: **Power spectrum density (PSD) analysis of the blending and Jigsaw augmentations.**

**Texture information analysis.** Our texture extraction module $T(x)$ is induced to lose the shape information so that we can ensure that the texture and semantic features are disentangled. To validate that $T(x)$ concentrates on the texture information, we analyze the power spectrum density (PSD) on the transformed CIFAR dataset (Figure B.1). For transformation of CIFAR, we blend the C10 and C100 in half ratios with alpha-blending (denoted as Blend). Also, we cut the image into four parts and randomly arrange them as Jigsaw augmentation (denoted as Jigsaw). We observe slight differences in the mid-level frequency for Jigsaw due to the hard crossing border at the center of the image but no significant change. Moreover, even though the sample's shape is mixed in the alpha-blend case, it can be seen that PSD is almost identical. These inspections support that $T(.)$ forgets the semantics (especially the shape clue) as expected when embedding the features.

**Semantic embedding results.** In Figure B.2, we plot the embedding results of the proposed semantics extraction module to verify the effectiveness of using semantic information. To visualize, we apply principal component analysis (PCA) to the extracted feature and make the dimension three. Figure B.2a depicts the scenario where MNIST is used as the ID dataset and other MNIST variants are the OOD. Figure B.2b shows the case where C10 is used as the ID dataset while C100 and SVHN are OOD. For both results, we observe that the embedding manifolds of the ID datasets are easily separable from the OODs' counterparts, and our semantics extraction module $S(x)$ concentrates on semantics information, such as labels of C10 and C100.

On the other hand, as presented in Figure B.2c and d, the semantics module $S(x)$ cannot identify the textural change such as the image distortions or resolution changes; all the embeddings lie in similar manifolds. These results are natural and expected since the semantics extraction module is designed to solely focus on the semantics side, not the texture.

**Power Spectrum Density (PSD) analysis.** To validate our assumption that texture information could be handled via Fourier transform, we visualize the PSD for all the datasets of the proposed

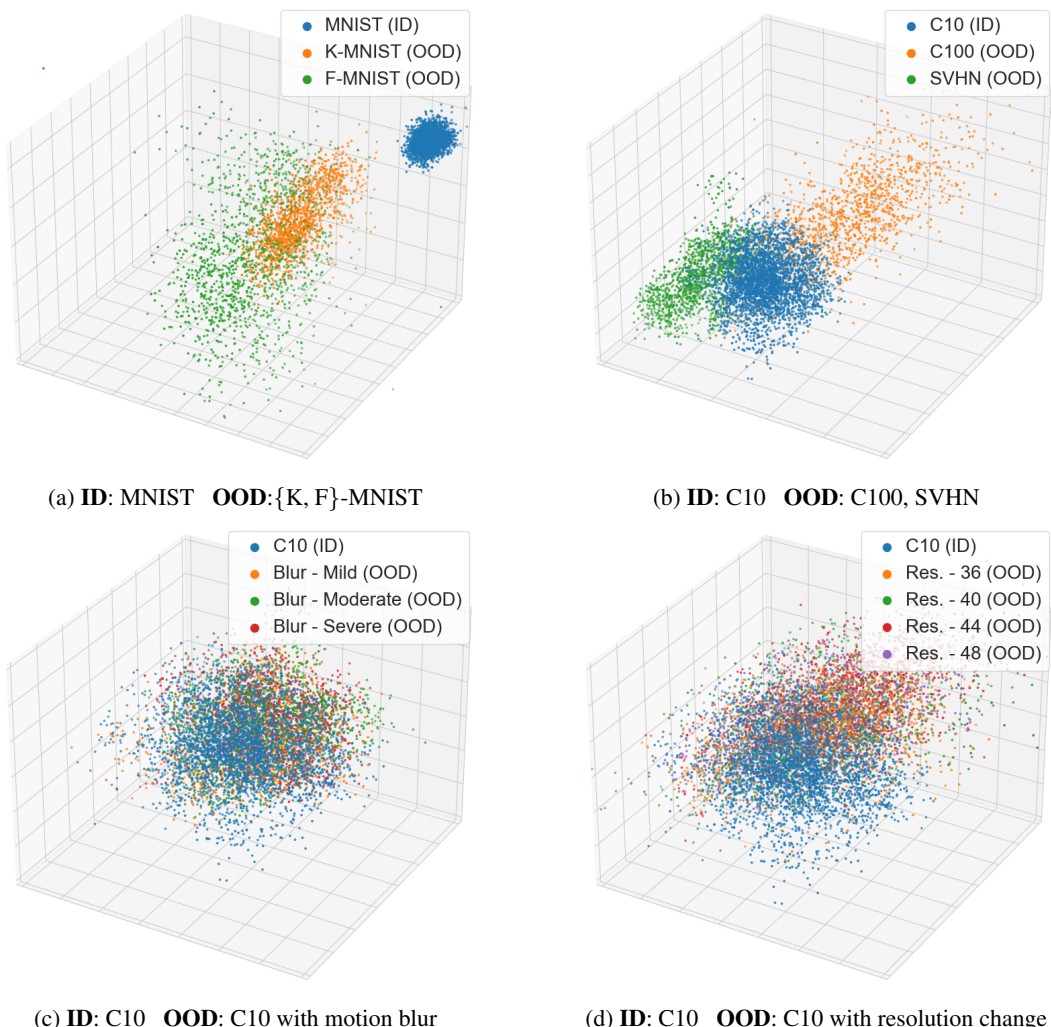

(a) **ID**: MNIST   **OOD**:{K, F}-MNIST

(b) **ID**: C10   **OOD**: C100, SVHN

(c) **ID**: C10   **OOD**: C10 with motion blur

(d) **ID**: C10   **OOD**: C10 with resolution change

Figure B.2: **Embedding results of the semantics extraction module.** The proposed semantics module $S(x)$ focuses on the semantics information. **(a, b)** The embedding outputs between ID and OOD are easily separable since the ID and OOD datasets have distinct semantic information. **(c, d)** It fails to disentangle the textural difference cases such as motion blur or image resolution change. This is a natural outcome since $S(x)$ is not guided to follow the textural cues.

OOD benchmark. In the case of the MNIST variant datasets (Figure B.3a), MNIST and K-MNIST show a similar spectrum because they have identical textures but only the semantics (letters) are different as illustrated in Figure A.1. On the other hand, F-MNIST has a different spectrum trend due to the disparate texture (letters vs. cloths). In Figure B.3b, we compare C10 to the C100 and SVHN datasets. Here, although C10 and C100 are separable in the embedding space (as shown in Figure B.2b), which has semantic information, the frequency domain-based approach, in which only texture information is extracted, cannot distinguish them. The SVHN dataset, however, shows disparate PSD, implying that the texture extraction module can identify this effectively.

We visualize PSD analysis results when the image distortions are applied to the C10 dataset (Figure B.3c-f). In many cases, the C10 dataset (blue) and mild corruptions (orange) appear as aligned spectrum, thus the texture-only mode may struggle in detecting mild corruptions as OOD. However, based on the real-world applications, we assumed that the mild distortion would be treated as ID, in other words, to be the textural robustness (Table 4). In contrast, severe corruptions (green) are distinctive compared to original C10, indicating that texture information is useful for detecting the severe corruptions as OOD.

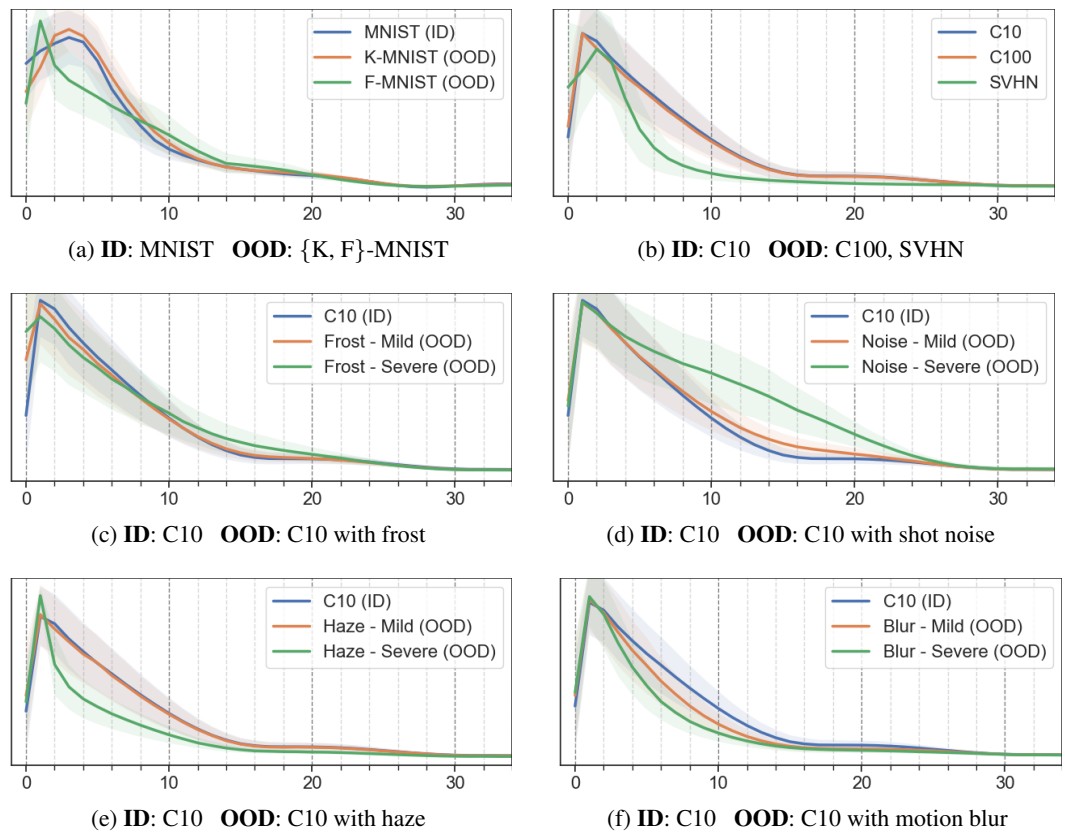

Figure B.3: **Power spectrum density (PSD) analysis.** We extend the PSD analysis conducted on Figure 4 to all the benchmark datasets we proposed.

## C    MODEL COMPARISON

Table C.1 presents the OOD detection performance of various methods when C10 and C100 datasets are set to the ID. We compare with R-Flow (Zisselman & Tamar, 2020), MOOD (Lin et al., 2021), 1-Dim (Zaeemzadeh et al., 2021) and G-ODIN (Hsu et al., 2020) algorithms. In the table, (c) and (r) depict the center-cropped and resized, respectively. To get a standard image resolution ($32\times32$), we downsize high-resolution datasets by described downsizing procedure. Note that the resized datasets are similar to the one presented in this work. We would like to emphasize that we mainly compare the competitors to our method with the $\lambda = 0.5$ setting. This makes the comparison fairer because $\lambda = 0.5$ does not require prior knowledge of which factor is important. Our method outperforms the previous algorithms in many cases and the proposed framework does not utilize any information other than the given ID dataset (C10 or C100) in the training phase. Unlike ours, the other methods use external information such as class labels or additional datasets to boost the performance.

| ID → OOD | | R-Flow | MOOD | 1-Dim | G-ODIN | Ours $\lambda =$ | | |
| | | | | | | 0.0 | 0.5 | 1.0 |
|---|---|---|---|---|---|---|---|---|
| C10 | SVHN | 98.2 | 96.4 | - | 98.8 | 86.1 | **99.9** | 99.9 |
| | TinyImgNet | 99.6 | - | - | - | 65.4 | **99.9** | 99.9 |
| | LSUN (c) | - | 99.2 | **99.4** | 98.3 | 68.3 | 90.9 | 98.6 |
| | LSUN (r) | **99.6** | 93.2 | 99.3 | 99.4 | 97.0 | 99.4 | 99.6 |
| | ImgNet (c) | - | - | 98.1 | **98.7** | 85.2 | 98.1 | 98.5 |
| | ImgNet (r) | - | - | 98.5 | **99.1** | 85.0 | 98.1 | 98.9 |
| C100 | SVHN | 95.1 | 85.8 | - | 95.9 | 80.9 | **99.9** | 100. |
| | TinyImgNet | 98.1 | - | - | - | 91.7 | **100.** | 100. |
| | LSUN (c) | - | **96.8** | 93.8 | 95.3 | 65.6 | 91.9 | 92.2 |
| | LSUN (r) | 98.9 | 77.6 | 95.7 | 98.7 | 97.0 | **99.8** | 99.6 |
| | ImgNet (c) | - | - | 88.6 | 97.6 | 94.1 | **100.** | 100. |
| | ImgNet (r) | - | - | 93.7 | **98.6** | 94.7 | 97.7 | 98.2 |

Table C.1: **Comparison on the conventional OOD detection benchmark.** We compare to other representative methods with various settings: R-Flow (Zisselman & Tamar, 2020), MOOD (Lin et al., 2021), 1-Dim (Zaeemzadeh et al., 2021) and G-ODIN (Hsu et al., 2020). Note that (c) and (r) indicate center-cropped and resized, respectively.

