# OpenReview forum: "Decomposing Texture and Semantics for Out-of-distribution Detection"
_ICLR.cc/2023/Conference — Submitted to ICLR 2023_

### Official Review · Reviewer_W1ZC · 2022-10-24

**Confidence:** 3
**Correctness:** 4
**Technical Novelty And Significance:** 2
**Empirical Novelty And Significance:** 4
**Recommendation:** 6

**Clarity, Quality, Novelty And Reproducibility:**

Clarity:

The paper is well written and clear. The figures are guiding the reader smoothly along the manuscript.

Section 3.3: why do you introduce polar coordinate to say afterward that you are not using those ? Then you mention f_r after equation (5) but it is not present in this equation. In the end, it is obscure to me what is T_i... Also T cannot belong to R^{2/w} and I don't think R should be indexed by w because it is the image size... Well, I think this section should be corrected...

Most citations should be in parentheses.

Quality:

Good.

The current framework and results might not be discussed enough. How does this work affects similar tasks such as anomaly detection, novelty detection, open set recognition or outlier detection ?


Novelty:

Their model is a combination of existing tools. The combination is original though.

Otherwise, I am not expert enough to judge beyond the authors claim.

Reproducibility:

I haven't seen any claim that the code will be released online upon acceptance.

**Strength And Weaknesses:**

Strength:
- a first attempt to clarify OOD detection task
- use of SOTA tools to design their models (resnet +SSVD for semantic, frequency features for textures and normalizing flow to combine both)
- achieve SOTA performances on different datasets
- ablation study is conducted

Weaknesses:
- I guess OOD detection is also meaningful for data that are not images. As such it seems only applicable to images.

**Summary Of The Paper:**

The authors introduces a new framework for OOD detection by defining more broadly the in-distribution (ID) samples using the notion of texture and semantic. Based on this definition, they create new benchmark scenarios. Finally, they propose a new method which perform well on both old and new benchmarks.

**Summary Of The Review:**

I would tend to recommend acceptance though I lack of expertise to have a strong opinion. The authors should really improve section 3.3. Apart from that my final decision will depend on other reviewers' opinion and the coming discussion.

---

> ### Author Response · Authors · 2022-11-17
> **Response to reviewer W1ZC**
>
> We are grateful that the reviewer found our method competitive and novel. We address comments in detail below:
>
> __Q1) Clarity on texture information technique.__
>
> A1) Thank you for your thoughtful comments. We described polar coordinates because this method inspired our proposed frequency information method from [1].
>
> In [1], the strength of the signal with respect to the radial radius band in the Fourier domain differs between real images and generated images by GAN-based methods. This ability to capture subtle perturbation motivates our texture extraction module.
>
> The notation f_r is not used later because we changed to cartesian coordinate of f_x, f_y. We assume that height and width are the same with the notation w.
>
> Also, we need to correct our T(x) formula, which may have confused the reviewer. We updated it in the revised version. The dimensionality of T(x) is R^w/2, not R^2/w. Therefore, the notation "i" from the T_i(x) can have a range of half the size of the image.
>
> __Q2) Most citations should be in parentheses.__
>
> A2) We apologize for the confusion. We corrected this in our revised manuscript.
>
>
> __Reference__
>
> [1] Dzanic, Tarik, Karan Shah, and Freddie Witherden. "Fourier spectrum discrepancies in deep network generated images." Advances in neural information processing systems 33 (2020): 3022-3032.

---

### Official Review · Reviewer_xaty · 2022-10-24

**Confidence:** 5
**Correctness:** 2
**Technical Novelty And Significance:** 3
**Empirical Novelty And Significance:** 3
**Recommendation:** 5

**Clarity, Quality, Novelty And Reproducibility:**

The paper is mostly well written. The disentanglement loss appears novel however it would be good to show its contribution for different lambdas.

I am not convinced that the "texture" branch is useful, especially since I can not see whether it could be useful in large images.

It is difficult to appreciate experiments from Tables 2-5 since they have not been tackled in the literature and some of them feel artificial. If the paper is accepted, I would propose moving them to the supplement and bringing Table C.1 to the main paper.



**Strength And Weaknesses:**

Strengths
1. Somewhat original approach to an important problem.
2. Competitive performance with respect to approaches that train with class labels (Table C.1) on very small images.
3. Initialization at the unit sphere and disentanglement loss appear to help (Table 6)

Weaknesses

1. \lambda=1.0 wins in almost all conventional OOD experiments (Table 1). The exception is C10-LSUN*. However, this result does not repeat in Table C.1 where \lambda=1 is better than \lambda=0 in both C10-LSUN configurations. If \lambda=1.0 always wins, then the contribution of the texture branch is unclear.
2. Table 6 does not reveal \lambda; it should be performed for all three values of \lambda
3. CIFAR-10 and CIFAR-100 have shared and related classes (dog - wolf, cat - tiger etc); this is not a suitable pair for OOD detection.
4. OOD detection on small images has been largely solved (Table C.1). Further work should evaluate on tougher benchmarks such as Species, SMIYC or FIshyscapes.
5. The distinction between anomalies and OOD samples does not reflect the current usage in the community. For instance, UCSD anomaly detection dataset [a] contains OOD objects. A recent review explicitly states that anomalies, outliers and novelties are synonyms but distinguishes low-level and high-level anomalies [b].
6. How does one choose K?

[a] Anomaly Detection in Crowded Scenes. CVPR 2010.
[b] A Unifying Review of Deep and Shallow Anomaly Detection. Proc. IEEE 2021.

Comments

The equation in 3.1 (λ · log p(T (x)|x) + (1 − λ) · log p(S(x)|x)) is different than (9).

Missing related work

The proposed approach appears related to [c]. The main difference is that [c] uses labels while the present approach appears to use a form of self-training.

Unit centers appear related to [d].

Detecting semantic OOD samples with a convolutional model is not easy due to texture bias [e,f]. It would be good to discuss how to deal with that problem (eg. [c]).

It would be sensible to discuss applicability for dense anomaly detection [g].

[c] Hybrid Models for Open Set Recognition. ECCV 2020.
[d] ArcFace: Additive Angular Margin Loss for Deep Face Recognition
[e] ImageNet-trained CNNs are biased towards texture. ICLR 2019.
[f] Why Normalizing Flows Fail to Detect Out-of-Distribution Data. NeurIPS 2020.
[g] SegmentMeIfYouCan - A Benchmark for Anomaly Segmentation. NeurIPS 2021.

**Summary Of The Paper:**

The manuscript proposes a novel approach for OOD detection based on two branches. One branch operates on DFT coefficients and the other on pixels. The pixel branch combines A ResNet-18 backbone, a multi-SVDD [ghafoori20icdm] layer initialized with unit centers, and a normalizing flow. The frequency branch is directly evaluated by a normalizing flow. The authors argue that the pixel branch is suitable for detecting semantic OODs, while the frequency branch is suitable for texture OODs.



**Summary Of The Review:**

As written in strengths/weaknesses, there is competitive experimental performance, interesting novelty (L_disentangle), but insufficient evidence about the contribution of the texture branch.

---

> ### Author Response · Authors · 2022-11-17
> **Response to reviewer xaty-Part 1**
>
> We thank the reviewer for their helpful comments and suggestions, which we address below:
>
> __Q1) Doubts about contribution of texture branch (lambda=0), and inconsistency found in optimal lambda configurations__
>
> A1) We appreciate the insightful discussion about our core idea and experiment settings.
> One might think that the texture module is not as valuable as the semantic one. However, the scenarios in conventional benchmarks are mostly focused on semantic OODs, for which our semantic module would work better than the texture module as expected.  Figure A.1(e) and (a) situations are essential illustrations of why lambda = 0 is necessary. Distortions and cracks are highly related to the textural clue. This discrepancy is necessary in many real-world applications, such as autonomous driving and industrial products inspection in factory conveyer belts.
>
>
> Regarding the result on C10-LSUN* case, we think that the experimental results differ because there is a strong correlation between image resolution and texture. In Table 1, we use 112x112 centrally cropped images from the high resolutions LSUN* example, which are taken from formal LSUN datasets. The LSUN examples from Table C.1, on the other hand, are datasets suggested by ODIN [1], a commonly used OOD detection study. This data was cropped or resized to a 32x32 size. Therefore, LSUN(c) may lose semantic information by cropping only the partial region of the whole image, and LSUN(r) may lose texture information.
>
> The image resolution results from Table 4 are also evident in this relation. That is, except for the base model ODIN, other models often detect OOD just by changing the resolution.
>
> We tried to develop a module that is independent to resolution change after observing this phenomena. Regarding this, we suggested the multi-SVDD variations for semantics module without using the ID class labels, and tried to decouple it from the texture counterpart.
>
> As seen in Table 5, while using semantic mode, our separation efforts have become less sensitive to modifications in texture and resolution.
>
>
>
>
> __Q2) Table 6 does not reveal Lambda__
>
> A2) Thanks for the suggestion. Table 6 analyzes the impact of our proposed methods in the semantic module, so the lambda setting is 1. During the rebuttal phase, we report additional results by varying lambda values as below.
>
>
>
> Table 1 Results of changing the lambda value in the ablation study
> | C10 -> C100 |        0         |  0.2  |        0.4 |        0.6 | 0.8 | 1.0   |
> |:--:|:--:|:--:|:--:|:--:|:--:|:--:|
> Multi-SVDD | 55.7     |       55.6   |   55.4     |  55.4    | 53.9  |      53.7
> \+ disentangle loss |  55.7      |       55.9   |   60.1    |   60.9     | 62.1     |    69.5
> \+ Angular init. |  55.7     |       63.3   |   75.5    |   88.8     | 91.1     |    93.5
>
> The results show that the performance of detecting semantic discrepancies (C10 -> C100) is affected by our semantic module changes.
>
>
> __Q3) CIFAR10 and CIFAR100 have related classes__
>
> A3) We appreciate the reviewer’s suggestion on this point. We think that three relevant previous works could be a reference for our experiment settings: [2] "Exploring the limits of out-of-distribution detection.", [3] "A simple fix to mahalanobis distance for improving near-ood detection."  [4] "Contrastive Training for Improved Out-of-Distribution Detection"
>
> The author of [4] claim that Near OOD is encountered more often in practice.
> Also by [2], the difficulty of the OOD detection task depends on how semantically close the outliers are to the inlier classes.
>
> However, we also agree with that the reviewer's concern, and this was our core idea of why using the lambda variable. Our proposed framework allows users to adjust the OOD definition. If we do not want to distinguish similar semantic information as suggested by the reviewer, setting lambda to 0 will detect it using only texture information.
>
> __Q4) OOD detection on small images__
>
> A4) Thank you for your experiment suggestion.
> We used the small images as the in-distribution because many prior works used these settings, such as ODIN.
> Especially , ODIN requires class label information for training, which limits the usage of datasets with no labels.
>
>   We provide additional experimental results on other datasets having larger sized images in the rebuttal.
>
>
>  Table 1 Using large images as the in-distribution dataset
>
> | ID Pets -> | Food | Flowers | CelebA |
> |:--:|:--:|:--:|:--:|
> |lambda = 0.0 	|	55.4 |		57.1	|	67.7 |
> |0.5 |	71.6 |		67.8 	|	75.3 |
> |1.0 	|	83.3 |		86.8 |		91.1 |
>
> The Pets [5], Food [6], and Flowers [7] datasets contain high-resolution images such as CelebA, which are larger than 112x112 in size initially. As expected from the OOD setting, our framework is most effective when semantic modules are activated. Further benchmarks considering other suggested datasets would be explored in our future studies.

---

> > ### Author Response · Authors · 2022-11-17
> > **Response to reviewer xaty-Part 2**
> >
> >
> > __Q5) Anomalies and OOD keyword usage__
> >
> > A5) Thank for your suggestion. Our work and terminologies used are based on ODIN and GRAM works [8, 9]. These studies define the OOD as samples that are drawn far away from in-distribution (i.e., distribution of training samples) statistically or adversarially.
> > Also, we concentrate on the "one-class" keywords, which differ slightly from [b] in terms of anomalies. In [1] and Deep-SVDD, the dataset is viewed as "In-distribution" or "Normal" in only one class, and the other classes are abnormal or OOD. In this regard, the OOD sample cannot be encountered in the training phase.
> >
> >
> > __Q6) How does one choose K?__
> >
> > A6) We choose K sufficiently larger than the usual number of classes in ID, for example, 512 in C10 from our case. These generated K seeds help to find appropriate data manifold distribution.   Figure 3 illustrates this assumption, and Figure B.2 shows the embedding results of our semantic module.
> >
> > __Q7) Equation__
> >
> > A7) We agree with your feedback. We revised this notation in our manuscript.
> >
> > __Q8) Related work__
> >
> > A8) We greatly appreciate your suggestions of other studies which are related to our main ideas. We have added these studies to our revised manuscript.
> >
> >
> > __Reference__
> >
> > [1] Shiyu Liang, Yixuan Li, and Rayadurgam Srikant. Enhancing the reliability of out-of-distribution image detection in neural networks. arXiv preprint arXiv:1706.02690, 2017
> > [2] Fort, Stanislav, Jie Ren, and Balaji Lakshminarayanan. "Exploring the limits of out-of-distribution detection." Advances in Neural Information Processing Systems 34 (2021): 7068-7081.
> > [3] Ren, Jie, et al. "A simple fix to mahalanobis distance for improving near-ood detection." arXiv preprint arXiv:2106.09022 (2021).
> > [4] Winkens, Jim, et al. "Contrastive training for improved out-of-distribution detection." arXiv preprint arXiv:2007.05566 (2020).
> > [5] https://www.robots.ox.ac.uk/~vgg/data/pets/
> > [6] https://www.kaggle.com/datasets/dansbecker/food-101
> > [7] https://www.robots.ox.ac.uk/~vgg/data/flowers/102/
> > [8] Shiyu Liang, Yixuan Li, and Rayadurgam Srikant. Enhancing the reliability of out-of-distribution image detection in neural networks. arXiv preprint arXiv:1706.02690, 2017.
> > [9] Kimin Lee, Kibok Lee, Honglak Lee, and Jinwoo Shin. A simple unified framework for detecting out-of-distribution samples and adversarial attacks. Advances in neural information processing systems, 31, 2018.

---

> > > ### Comment · Reviewer_xaty · 2022-11-21
> > > **Response to the rebuttal**
> > >
> > > I have re-read the reviews and the author responses. The main weakness remains: the experiments do not support the utility of the Fourier branch. Consequently, I keep my pre-rebuttal rating.
> > >
> > > Earlier research shows that normalizing flows can not detect anomalies in images, but only in discriminative features [kirichenko20nips]. This manuscript shows that training with Multi-SVDD (plus angular initialization and disentanglement) can compensate for the lack of labels. Focusing on this effect would likely make this manuscript much better.
> > >
> > > If this paper is rejected, I propose the following revisions:
> > > - ablate FFT (frequency components may not contribute, as also noted by UnLe)
> > > - ablate Multi-SVDD
> > > - perform experiments on MVTec (this is a good match since due to having no labels)

---

### Official Review · Reviewer_UnLe · 2022-10-24

**Confidence:** 3
**Correctness:** 2
**Technical Novelty And Significance:** 2
**Empirical Novelty And Significance:** 2
**Recommendation:** 5

**Clarity, Quality, Novelty And Reproducibility:**

The paper is clearly written and the main ideas are effectively presented. The method presented in the paper is new, however, I am not sure of it's efficacy.


The codes are not provided with the paper.

**Strength And Weaknesses:**

The main strenghts of the paper are as follows:

1) The paper is well-written and the main ideas are presented in a succinct manner.
2) A problem of interest to the community is tackled in the paper, where better definition of in-distribution and OOD data will allow for deployment of deep learning based solutions in real-life scenarios.


The main weakness in my opinion is as follows:
1) I don't believe frequency based mehtods are a good proxy for texture of a region. In fact most texture segmentation methods I am aware of don't use frequency based features.
2) In principle if frequency components are indeed important for the OOD data, we should be able to learn it with a neural network. FFTs are just scaled dot product of the input with FFT basis.
3) I am not totally sold on the distinciton of the semantics and the textures. How do we decouple the texture from the semantics. I did not get it from the paper.

**Summary Of The Paper:**

The paper presents a definition of the in distribution data (and by extension OOD), where the authors decompose the definition of the ID into texture and semantics. This decompostion provides the flexibility to different scenarios by determining which view of the ID more suited for a given scenario. The texture pipeline is built form 2D FFT and the semantic component is build from ResNet and multi-SVDD.

**Summary Of The Review:**

I am not totally sold on the efficacy of the method and I dont agree with the main thesis of the paper, i.e., separation of the texture and the semantic component of the images and use it to identify in and out of distribution data. As such I will lean toward rejecting the paper.

---

> ### Author Response · Authors · 2022-11-17
> **Response to reviewer UnLe**
>
> We are thankful that the reviewer found our viewpoint of OOD interesting. We address the comments in detail below:
>
> __Q1) Doubts about frequency-based features for texture information.__
>
> A1) As the reviewer commented, for texture segmentation tasks, we think the Fourier transform-based methods are not very suitable for extracting "local" information because the outputs of the Fourier transform represent the entire image's information.
>
>
> However, we claim that the frequency features can provide useful information to our target task. We think that the following works support our argument.
>
> [1] Frank, Joel, et al. "Leveraging frequency analysis for deep fake image recognition." International conference on machine learning. PMLR, 2020.
> [2] MLAEl Helou, Majed, Ruofan Zhou, and Sabine Süsstrunk. "Stochastic frequency masking to improve super-resolution and denoising networks." European Conference on Computer Vision. Springer, Cham, 2020.
> [3] Yin, Dong, et al. "A fourier perspective on model robustness in computer vision." Advances in Neural Information Processing Systems 32 (2019).
>
> [1] solve the deep fake image detection using the Fourier power spectrum. They found that deep fake images have artifacts in the frequency domain.
>
> [2] propose a contrastive learning-based method for enhancing denoising quality, which is related to our texture-guided benchmark. They cut off the images in the frequency domain in the training phase.
>
> The image classification can obtain high performance by relying on local statistics that are correlated with texture. [3] claim that high-frequency patterns can be exploited in a way to achieve notable i.i.d generalization. They test base models on ImageNet data when severe filtering is applied to the input in the frequency domain.
>
> Figures 4 & B.1 also illustrate the frequency information impacts on texture skim. As shown in Figure 4, the CIFAR10 and CIFAR100 have similar power spectrums, but corrupted CIFAR10 does not. Moreover, Figure B.1 shows that only semantically destroyed images also have a similar pattern in the frequency domain.
>
>
> __Q2) Frequency features need to be learned from a neural network.__
>
> A2) Flow-based deep learning model explicitly learns log-likelihood depending on inputs.
> In this study, we feed the Fourier features into the flow-based texture model. We can compute texture log-likelihood using learned texture extraction parameters from the Flow-based model. Furthermore, rather than just using DFT results, we allowed only texture information to be extracted from the Fourier transform results by losing the direction information from DFT.
>
> __Q3) Decoupling two factors.__
>
> A3) We agree with the reviewer's opinion in part. Recently, many vision studies have been trying to decouple intrinsic properties from images, but it is still challenging because many properties are entangled in dataset manifolds. However, we show that OOD detection performance can be greatly improved by separately considering at least two intrinsic properties, which have not been considered in previous OOD detection studies. We have tried to isolate the texture and shape as much as possible by using a flow-based model that explicitly learns log-likelihood and by employing a disentanglement loss.  Table 6 shows the impact of our disentanglement method.
> We claim that these efforts will be more beneficial for the complex problems presented in section 4.2.
>
>
> __Reference__
>
> [1] Tarik Dzanic, Karan Shah, and Freddie Witherden. Fourier spectrum discrepancies in deep network generated images. arXiv preprint arXiv:1911.06465, 2019.
> [2] MLAEl Helou, Majed, Ruofan Zhou, and Sabine Süsstrunk. "Stochastic frequency masking to improve super-resolution and denoising networks." European Conference on Computer Vision. Springer, Cham, 2020.
> [3] Yin, Dong, et al. "A fourier perspective on model robustness in computer vision." Advances in Neural Information Processing Systems 32 (2019).

---

### Official Review · Reviewer_BBXZ · 2022-10-27

**Confidence:** 3
**Correctness:** 3
**Technical Novelty And Significance:** 2
**Empirical Novelty And Significance:** 3
**Recommendation:** 6

**Clarity, Quality, Novelty And Reproducibility:**

The explicit separation of high and low level features seems to be novel and enables a specific architecture. The new benchmark data set is a novel contribution.

**Strength And Weaknesses:**

The overall analysis of the problem is interesting and provides a justification for the design choices and dual-pathway classifier. The angular projection of the training data to enhance OOD detection is also an interesting approach to capturing the concept of the training set boundaries.

The use of rectangular frequency binning, and discarding orientation, as a texture descriptor is very rough. Empirically, it seems to work well enough, but a more sophisticated set of features may improve performance. Does the process make use of an integral image to do the computation of rectangular sums? It's also not clear if the computation is being done on the base DFT or the power spectrum. The text for figure 4 implies it is the power spectrum.

The acronym SVDD does not seem to be defined in the text.

The balance of the two systems is evaluated at only lambda = {0, 0.5, and 1}.  It's not clear from the results that equal weighting will necessarily be the best option.

The new benchmark data set seems to be needed, given the performance on the prior data sets. It seems tailored to demonstrating this particular differentiation of the task (high level versus low level features).  My question is whether it has broader applicability.

**Summary Of The Paper:**

The paper outlines an approach to out of distribution [OOD] example identification. The approach makes use of the idea of explicitly separating high-level semantic content and low-level image frequency content and training separate classifiers whose outputs are combined to make the OOD decision. The semantic content extraction network uses a novel approach to data pre-processing intended to make it easier to identify OOD elements.

**Summary Of The Review:**

The conceptual approach leads to a specific architecture that shows good performance on both the existing data sets and a new proposed benchmark data set. The concept of thinking about OOD detection as having separable problem characteristics enables new approaches to the problem.

---

> ### Author Response · Authors · 2022-11-17
> **Response to reviewer BBXZ**
>
> We thank the reviewer for the constructive comments and suggestions, which we address below:
>
> __Q1) Discussion on the proposed frequency binning methods.__
>
> A1) Thank you for your suggestion. As the reviewer asked, we used rectangular sum using the DFT basis because this method requires lower time complexity than the prior work [1] without a performance drop.
>
>
> Our simple rectangle sum method does not necessitate a polar coordinate transform, in contrast to Equation (4) from [1].
> We compare the two methods' preprocessing times and texture module performance in this rebuttal.
>
>
>
>  Table 1 Time complexity and performance comparison with the polar coordinate-based method in [1]
> |Method |        Preprocessing Time (C10)         | Preprocessing Time (CelebA)  |        Texture module OOD result C10 -> SVHN  |
> |:--:|:--:|:--:|:--:|
> Polar | 25 sec     |      953 sec   |   85.9
> \Rectangular |  19 sec    |      631 sec  |   86.1
>
>
> These results show that our methods have lower time complexity with no performance degradation. However, as the reviewer suggested, utilizing more complex texture descriptors in an efficient manner would also be an important focus of our future research.
>
> When we analyze our results in contrast, we used power spectrum density instead of DFT. This may have caused the confusion. Figure 4 uses the PSD method to demonstrate how the texture aspects in the Fourier domain display a certain pattern between corrupted C10 and C10.
>
> __Q2) The acronym SVDD does not seem to be defined in the text.__
>
> A2) We apologize for missing it out. We revised this in our manuscript
>
> __Q3) Limited lambda configuration__
>
> A3) We agree with the reviewer's concern that setting equal weighting is not necessarily optimal in all cases. Nonetheless, we evaluate lambda = 0.5 settings for the cases where there is no prior knowledge of whether semantic or textural OODs should be considered.
>
> We provide the results of additional experiments performed with lambda values increased by 0.2.
>
>
> Table 1 Changing lambda values.
> |ID -> OOD |        lambda = 0         | 0.2  |        0.4  |        0.6  | 0.8  | 1.  |
> |:----:|:--:|:--:|:--:|:--:|:--:|:--:|
> BTAD Normal -> Anomaly |92.1 | 92.0 | 91.5 |84.0 |73.5 | 61.0
> Rotate MNIST 0 angle -> 75 |   59.1 | 71.5 | 90.4 | 91.9 | 96.9 |99.8 |
>
> These results show that if one of the information from our assumption overwhelms the other, then we have to decide the lambda settings by considering application situations. Please note that lambda values near 0.5 also produce reasonable performance (The results of lambda=0.5 settings are 88.8, 90.0, respectively).
>
>
>
> __Q4) Broader applicability__
>
> A4) Thank you for your valuable comment. In this work, we focus on decomposition of two major factors in images for OOD detection, which have not yet been considered in recent OOD detection studies.
>
> We also argue that if we can find another key properties and descriptors of images, they can be easily incorporated into our framework using Equations 8 and 9. For example, using an extractor of the given information, such as T(x), we can learn it in the Flow-based model to obtain each log-likelihood, such as logP(T(x)|x). After that, these multiple modules can be learned as independently as possible by the disentanglement loss in equation 8 and fusing them to our framework using equation 9.
>
> We also want to underline that we have experimented more with different textures and semantics settings, which are shown in reviewer xaty's response, for further performance validation under various settings.
>
>
>
>
>
>
> __Reference__
>
> [1] Tarik Dzanic, Karan Shah, and Freddie Witherden. Fourier spectrum discrepancies in deep network generated images. arXiv preprint arXiv:1911.06465, 2019

---

### Decision · Program_Chairs · 2023-01-20

**Decision:**

Reject

**Justification For Why Not Higher Score:**

There was concern about the usefulness of the texture branch. Without that branch, the proposed approach loses its appeal.

**Justification For Why Not Lower Score:**

N/A

**Metareview: Summary, Strengths And Weaknesses:**

The paper proposes a new approach to OOD dectection based on both DFT coefficients and pixels.

All reviewers found the idea interesting. However, there were concerns about the usefulness of the texture branch raised by two reviewers (UnLe has an issue with using frequency based methods to characterize the texture while xaty was concerned about the good performance of lambda=1).

After further discussion with the reviewers about this issue, the conclusion was reached that this was too limiting and as such I recommend rejection. I encourage the authors to leverage all the comments made for a future resubmission.

**Summary Of Ac-Reviewer Meeting:**

There was no video meeting but a further discussion with reviewers on openreview. As can be seen, the two reviewers who gave a 6 did not fight for acceptance. I am also wary of novel perspectives that end up not being too useful and this seemed to be the case here.